# Evaluation of a Novel Three-Dimensional Robotic Digital Microscope (Aeos) in Neurosurgery

**DOI:** 10.3390/cancers13174273

**Published:** 2021-08-25

**Authors:** Stefanie Maurer, Vincent Prinz, Lina-Elisabeth Qasem, Kristin Elizabeth Lucia, Judith Rösler, Thomas Picht, Jürgen Konczalla, Marcus Czabanka

**Affiliations:** 1Department of Neurosurgery, University Hospital Frankfurt, 60528 Frankfurt am Main, Germany; Stefanie.Maurer@kgu.de (S.M.); Vincent.Prinz@kgu.de (V.P.); Lina.Qasem@kgu.de (L.-E.Q.); Kristin.Lucia@kgu.de (K.E.L.); Juergen.Konczalla@kgu.de (J.K.); 2Department of Neurosurgery, Charité-Universitätsmedizin Berlin, 10117 Berlin, Germany; Judith.Roesler@charite.de (J.R.); Thomas.Picht@charite.de (T.P.)

**Keywords:** Exoscope, operating microscope, microsurgery, ergonomics

## Abstract

**Simple Summary:**

Current literature debates the development and implementation of three-dimensional Exoscopes in the daily routine of neurosurgical practice. This study evaluates the grade of satisfaction and intraoperative handling of the novel Aesculap Aeos Three-Dimensional Robotic Digital Microscope used in a Neurosurgical Department in the daily practice in a larger series of neurosurgical procedures. Our evaluation of this modern microscope technology focuses on three central aspects of microsurgical procedures. First, the visualization of tumor tissue and its discernment from healthy tissue, which is of utmost importance during neurooncological procedures. Furthermore, the aspect of ergonomics and fatigue during long, repetitive surgical procedures have been shown to cause work-related musculoskeletal disorders. Finally, the implementation of novel three-dimensional microscopes may be a promising tool for surgical education. Improved image quality, superior ergonomic comfort for the surgeon and increased accessibility within the surgical field have been noted as major advantages to date.

**Abstract:**

Objective: Current literature debates the role of newly developed three-dimensional (3D) Exoscopes in the daily routine of neurosurgical practice. So far, only a small number of cadaver lab studies or case reports have examined the novel Aesculap Aeos Three-Dimensional Robotic Digital Microscope. This study aims to evaluate the grade of satisfaction and intraoperative handling of this novel system in neurosurgery. Methods: Nineteen neurosurgical procedures (12 cranial, 6 spinal and 1 peripheral nerve) performed over 9 weeks using the Aeos were analyzed. Ten neurosurgeons of varying levels of training were included after undergoing device instruction and training. Following every surgery, a questionnaire consisting of 43 items concerning intraoperative handling was completed. The questionnaires were analyzed using descriptive statistics. Results: No intraoperative complications occurred. Surgical satisfaction was ranked high (78.95%). In total, 84.21% evaluated surgical ergonomics as satisfactory, while 78.95% of the surgeons would like to use this system frequently. Image quality, independent working zoom function and depth of field were perceived as suboptimal by several neurosurgeons. Conclusion: The use of Aeos is feasible and safe in microsurgical procedures, and surgical satisfaction was ranked high among most neurosurgeons in our study. The system might offer advanced ergonomic conditions in comparison to conventional ocular-based microscopes.

## 1. Introduction

The development and implementation of three-dimensional (3D) Exoscopes in the daily routine of neurosurgical practice is a matter of current debate [1,2,3,4,5]. Superior ergonomic comfort, improved image quality and magnification, increased accessibility of the surgical field and enhanced experience for residents during the surgical procedure are noted as the main advantages versus the use of the established standard operating microscope (OPMI) [1,6,7,8]. Instrument handling and intraoperative repositioning were highly rated by surgeons after having performed cranial and spinal procedures with an Exoscope [9]. In contrast, a lack of stereoscopic vision, the higher time requirement for changing the depth and focus of the operating site and the perception of depth were ranked lower compared to the use of OPMI [7,10,11].

So far, the majority of reports have focused on the VITOM (Karl Storz, Tuttlingen, Germany), the Olympus ORBEYE (Olympus, Tokyo, Japan) and the Zeiss KINEVO (Carl Zeiss AG, Oberkochen, Germany) (Table 1). Only a small number of experimental setups, a cadaver lab study or case reports referred to the utilization of the novel Aesculap Aeos Three-Dimensional Robotic Digital Microscope [10,12,13].

Therefore, the aim of this study was to evaluate the grade of satisfaction and the intraoperative handling of a newly developed Three-Dimensional Robotic Digital Microscope Aeos used in a Neurosurgical Department in the daily routine in a larger series of surgical procedures.

## 2. Material and Methods

### 2.1. Study Design

In this prospective study, 19 neurosurgical procedures were performed using the Aeos Three-Dimensional Robotic Digital Microscope over a period of nine weeks in a German neurosurgical university department. A cohort of ten neurosurgeons, including the head of the department, four senior physicians, two consultants and three residents scheduled as the headsurgeon for a procedure corresponding to their level of training (Figure 1). The procedures analyzed included cranial neurooncological operations, spine procedures, epilepsy surgery, aneurysm clipping, cranial trauma surgery and peripheral nerve surgery.

Every surgeon underwent a formal introduction to the Robotic Digital Microscope from the manufacturer prior to first use. During surgery, the surgeon was able to change to the conventional microscope at any step of the procedure as deemed necessary by the surgeon. Only procedures deemed suitable for exoscopic surgery were included to be performed with the Aeos.

Patient age, diagnosis, surgical procedure, histopathological findings, case complexity as well as surgical time and clinical outcome were documented. In addition, a questionnaire consisting of 43 items using a 5 (to 6) -point Likert scale regarding the intraoperative satisfaction in terms of image quality, ergonomics, ease of use and physical exhaustion afterwards was completed after every operative procedure.

### 2.2. Technical Specifications and Features of the Three-Dimensional Robotic Digital Microscope

The Aesculap Aeos Three-Dimensional Robotic Digital Microscope is a robotic-assisted 3D Heads-Up Surgery System. It consists of a camera, a 6-axis robotic arm for flexible setup options, a 3D surgical screen (16:9 wide view) with HDR imaging (Full HD (1080p HD stereoscopic image) or 4K UHD) and 3D glasses, a control screen (15.6″ display size) with a touchscreen, the base of the microscope (3D recording, video outputs, video inputs (MRI or CT scans), USB, DICOM,) and a footswitch (wireless or cabled, programmable buttons, joystick) (Figure 2). The system promises a superior depth of field, in addition to a wider and cooler homogenous field, due to direct coaxial LED illumination (white light and fluorescence). The Robotic Microscope offers a 10× optical zoom with a working distance of 200–450 mm. Backlight illuminated 3 ICG (DIR 800) fluorescence, including the playback function with slow motion for analyzing perfusion and the 3D 5-ALA (DUV 400) fluorescence for malignant gliomas. The integration of white light along with fluorescence is available for the use of resections performed with 5-ALA. Positioning of the robotic arm is performed via handles or footswitch. The lock on target function allows for observation of a fixed point in the operative field from different angles. Setting Waypoints during the procedure also gives the opportunity to save camera positions and to return to them at a later time.

### 2.3. Statistical Analysis

Statistical analysis was performed using IBM SPSS Statistics Version 26. A total of 43 questions were evaluated and presented as mean and standard deviation (mean (SD)).

## 3. Results

### 3.1. Surgical Procedures and Patient Characteristics

A total of 19 operative procedures in 19 different patients were performed fully or partially with the Aeos Three-Dimensional Robotic Digital Microscope by ten different neurosurgeons, including the head of the department, four senior physicians, two consultants and three residents. Twelve cranial procedures were performed, including three glioblastomas (parietotemporal, temporal and frontal) (Figure 3), three metastases (two cerebellar, one parietooccipital), two meningiomas (convexity frontal, sphenoid wing) as well as two vascular cases with clipping of an internal carotid artery and medial cerebral artery aneurysm (Table 2, Figure 4). One trauma case with the tube-assisted evacuation of an intracerebral hemorrhage in the basal ganglia was conducted (Figure 5). In total, six spinal cases were addressed including one meningioma TS 4/5 (Figure 1), one hemangioblastoma LS 3/4 (Figure 6), two spinal canal stenosis (LS 2/3, CS 5-7, both fusion procedures) and two lumbar disc herniation L5/S1. One epilepsy-related surgery and one vagal nerve stimulation (Figure 7) were evaluated additionally. The different localizations, procedure types or diagnosis are presented in Table 2. 

The mean age of the patients was 59.3 years (range 26–85 years, ±17.3 years). Case complexity was rated high in 10.53%, of increased difficulty in 36.84%, medium in 42.11% and relatively low in 10.53% of the cases (*n* = 19). No procedure was rated as low or very high in terms of the case complexity. 

The mean time in the operation theatre was 221 min (±71 min), the mean surgical time was 154 min (±61.5 min). Gross total resection (verified by MRI) of the three glioblastomas was achieved in two cases, and the third one was planned as a subtotal tumor debulking. 

One postoperative complication (trochlear nerve palsy) after temporal- and amygdala resection occurred. No intraoperative complications were detected. Concerning the patients’ outcome, seven patients improved postoperatively (pain decreased in four cases, motor strength impairment improved in three cases).

### 3.2. Questionnaire Evaluation

#### 3.2.1. Applicability of the System

A total of 78.95% of the surgeons rated the surgical satisfaction as high after having performed the operation, and 26.32% evaluated the satisfaction as very high. None of the 19 surgeons reported an unsatisfactory surgical experience. A total of 68.42% of surgeons agreed that the case was suitable for operating using the Aeos microscope (Figure 8). Both vascular procedures (aneurysm clipping) were assessed as not (easy) suitable for operating with the robotic digital microscope (Figure 4). 

A total of 84.21% of participants evaluated the surgical ergonomics as satisfactory. 

Overall case complexity was rated as high in two cases (10.53%) (vascular procedures), increased in seven cases (36.84%), medium for eight cases (42.11%) and relatively low in two cases (10.53%) (cerebellar metastases). 

When asked whether the microsurgical tasks required during the case were suitable for the Aeos microscope, 63.16% agreed or strongly agreed (36.84%). 

Controlling the 6-axis robotic during the procedure was easy for 52.63% of the surgeons, whereas 21.05% found the handling difficult. The hand-eye coordination was affected negatively during the digital robotic microscope utilization in two out of 19 times. 

The overall 3D depth perception was satisfactory in 73.68% of the cases and unsatisfactory in 21.05% in terms of image quality. The sharpness of the displayed picture on the screen was rated as satisfactory in 94.74%.

A total of 78.95% assessed the depth of the field as satisfactory, while 10.53% disagreed. The luminance was found to be satisfactory in 94.74% of cases.

#### 3.2.2. Usability of the System

A total of 78.95% of the surgeons in the current study would like to use the Aeos Microscope more frequently. None rated it as unnecessarily complex. Twelve participants indicated that the microscope was easy to use (63.16%), ten felt confident using it alone (52.63%), six would need the support of a technical person if using it frequently (31.58%), and only five surgeons answered that the system was cumbersome in use (26.32%). Out of the surgeons, 89.74% found that most people could learn to use the Aeos Microscope very quickly (Figure 9).

#### 3.2.3. Workload of the Performing Surgeon

When asked how mentally fatiguing the procedure was, the cognitive and physical demands were rated as low (Figure 10).

Furthermore, task complexity was rated as medium and values for situational stress and distractions were rated as low. 

### 3.3. Younger vs. Experienced Surgeons

When asked about surgical satisfaction after using the new device, residents rated it as neutral to high, senior physicians as high to very high.

The senior physicians, as well as the residents, evaluated the physical and mental tiredness after the operation as very low to medium. 

The control of the 6-axis robotic arm was evaluated as rather easy by the residents and the experienced surgeons, the surgical working zone as unblocked by the consultants and senior physicians and rather restricted by one experienced surgeon. 

Surgical ergonomics, in general, were highly satisfactory and valued by the senior physicians. On the other hand, the residents rated the ergonomics as neutral. In terms of the 3D depth perception, the majority of the surgeons were satisfied to very satisfied. Residents agreed on wanting the support of a technical person during periodic use of the system, but the consultants and the majority of the senior physicians disagreed in that matter.

## 4. Discussion

The evaluation of modern three-dimensional robotic digital microscope technology highlights three central aspects of modern microsurgical procedures.

First, the visualization of tumor tissue and its discernment from healthy tissue is of utmost importance during neurooncological procedures. The introduction of 5-ALA-induced fluorescence has been a major scientific achievement in the intraoperative visualization of malignant tissue, which is independent of brain shift [14,15,16,17,18,19,20,21]. In conventional microscopic surgery, tumor tissue “glows” after preoperatively administered 5-ALA and intraoperative switching from conventional white light to a blue light source, which is integrated into the microscope. However, due to the dimming of white light, healthy, non-fluorescent brain tissue is barely visible in conventional microscopy, as illustrated in Figure 3. 

The Three-Dimensional Robotic Digital Microscope combines the utilization of blue light and white light during the resection of high-grade malignant tumors using 5-ALA-induced fluorescence. Consequently, non-fluorescent brain tissue appears more visible next to fluorescent tumor tissue, and tumor’s margins become even more definable. 

Furthermore, the aspect of ergonomics and fatigue during long, repetitive surgical procedures have been shown to cause work-related musculoskeletal disorders (WMSDs).

Gadjradj et al. [22,23,24,25] report that 73.6% of 417 neurosurgeons had experienced WMSDs, 11.3% had to take time off work, and even 14.2% thought about changing specialties or their career because of the work-related musculoskeletal pain. 

Mavrovounis et al. showed even more concerning results after having interviewed 409 members of the “European Association of Neurosurgical Societies”, the “Neurosurgery Research Listserv” and the “Latin American Federation of Neurosurgical Societies” [26]. Out of the 409 members, 87.9% had experienced WMSDs most commonly described as neck and shoulder pain. 

In general, neurosurgeons have become accustomed to restricted intraoperative working conditions causing impaired ergonomics during surgery.

Other drawbacks concerning working with a conventional microscope are the weight, its size, short depth-of-field (at high magnification) and the ocular dependence [27].

The newly developed Three-Dimensional Robotic Digital Microscope, as well as other Exoscopes, now promise superior ergonomics posturing while operating.

In our study, the feasibility of use in daily neurosurgical practice appears high. 

Both the younger as well as the more experienced surgeons agreed on the good usability of this system for most applications. The surgical satisfaction after having performed a procedure with the Aeos was high in 78.95% of all cases. This high level of contentment is comparable or even better than described in the current literature [1,5,6,28]. The rate of complications both intra- and postoperatively following exoscopic surgery was low. These findings are in line with previous studies [1,2,4,5,29].

In our series, two cases of aneurysm clippings were rated as not (easy) suitable for the use of the Aeos, and the surgeon switched to the conventional microscope. These results differ from previous studies concerning the role of exoscopic surgery in the operative treatment of vascular cases [2,30], illustrating the need for larger studies that may elucidate predictive factors for the suitability of cases using exoscopic versus conventional microscopic techniques. In contrast, we felt safe to clip the aneurysm, but the different positioning was uncomfortable. Assuming that a pterional MCA and ICA-aneurysm clipping is one of the most standardized procedures in neurosurgery (instead of tumors, which were located differently), we have to focus on preparing the Sylvian fissure, which was uncomfortable due to the different positioning. Therefore, we would recommend having a conventional microscope next to an exoscope and training to use the exoscope. We used it in our fifth case and handled it well but not as fast as with conventional microscopes, which differs from the experience in other procedures. Figure 4A,C demonstrates the ICG mode of the Aeos system during the clipping procedure. For even better reference Figure 4D,F shows even enhanced contrast by image processing, which during the procedure is not necessary due to intraoperative high definition backlighted LED screen.

Overall, the current state of research on the emerging field is sparse due to the novelty of the exoscopic or three-dimensional digital robotic technique. Working with the familiar conventional microscope on which every neurosurgeon is trained since the beginning of his career is still the gold standard in neurosurgical departments. 

As such, in our study, the experienced surgeons, in particular, described the new device, different working angles, the utilization of the 6-axis robotic arm and focusing on the screen as irritating at the beginning. 

The implementation of the novel Three-Dimensional Robotic Digital Microscope into the daily routine of a neurosurgical department may also be a promising tool in terms of surgical education. The lead surgeon, as well as the assistant, are able to see the whole procedure in a 3D high-resolution image on a large screen. The assistant watches all steps at exactly the same angle as the operating surgeon in an equally ergonomic position [13]. The learning and teaching opportunities in this context are encouraging. 

A number of limitations or disadvantages of exoscopic surgery have been noted. In general, the default settings for image quality and depth of field were perceived as suboptimal by some of the participating neurosurgeons. Blood vessels were displayed in an overly bold red, and the visualization of hemorrhagic tissue was, therefore, a subjective limitation as previously reported [3]. This might be due to a lack of routine of the surgeons in adjusting and working with the new device and control system. Some surgeons pointed out that the independent working zoom function, as well as an intermittently blocked view, was irritating at the beginning. These circumstances could be addressed by individualizing personal software settings in the system. Another limitation described in the literature is headaches occurring after working with a 3D exoscope [3]. During our study set up, the majority of surgeons rated the procedure as only minorly physically or mentally fatiguing. 

## 5. Limitations

One possible limitation of this study could be the number of different operating surgeons (*n* = 10) at different educational levels, which does not allow for statistically comparable cohorts. Furthermore, learning curves and progress reports over time were not possible due to the inclusion period of nine weeks. Another limitation may be the diversity of included cases with elective cranial, spinal and peripheral nerve surgery as well as trauma surgery. 

## 6. Conclusions

In conclusion, the Aeos Three-Dimensional Robotic Digital Microscope seems feasible for safe utilization in a wide spectrum of microsurgical procedures in neurosurgery. Surgical satisfaction was ranked high among the majority of neurosurgeons in our study. The system might offer advanced ergonomic conditions in comparison to conventional ocular-based microscopes.

## Figures and Tables

**Figure 1 cancers-13-04273-f001:**
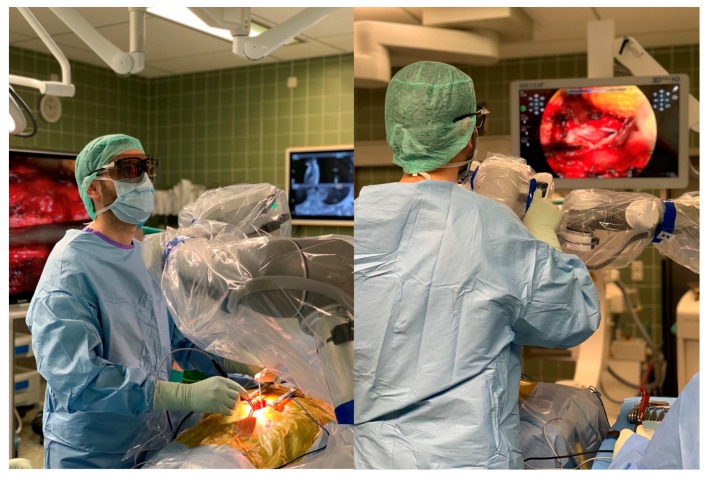
The surgeon during an operative resection of a meningioma TS 4/5 using the Three-Dimensional Robotic Digital Microscope.

**Figure 2 cancers-13-04273-f002:**
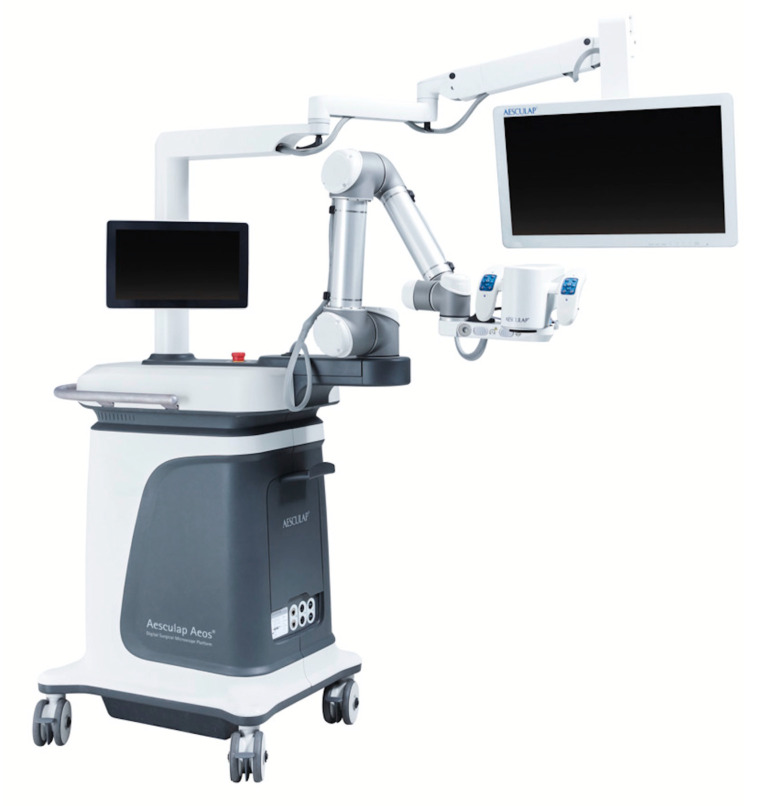
The Aesculap Aeos Three-Dimensional Robotic Digital Microscope.

**Figure 3 cancers-13-04273-f003:**
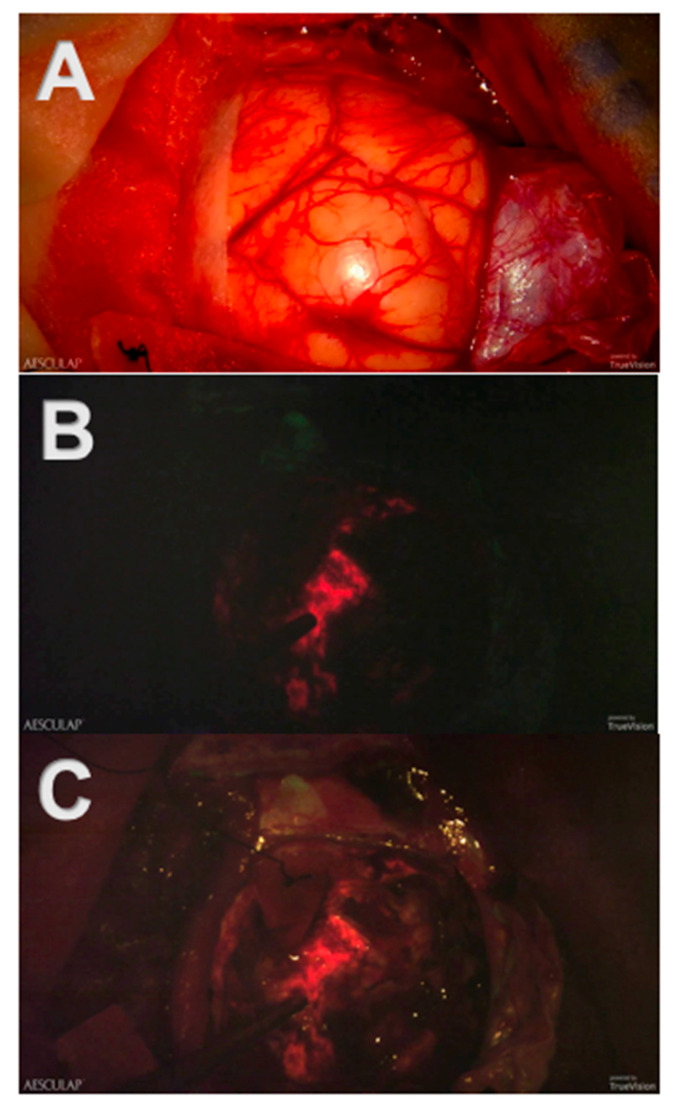
Screenshots of intraoperative recordings with Aeos during resection of a glioblastoma multiforme assisted by 5-ALA-induced PpIX fluorescence: (**A**) Conventional white light. The tumor and its margins are elusive under conventional white light. (**B**) Blue light. The tumor is clearly visible in strong fluorescence under dimmed white light and activated blue light. (**C**) Simultaneous usage of the white and blue light source: Tumor margins are easier to identify.

**Figure 4 cancers-13-04273-f004:**
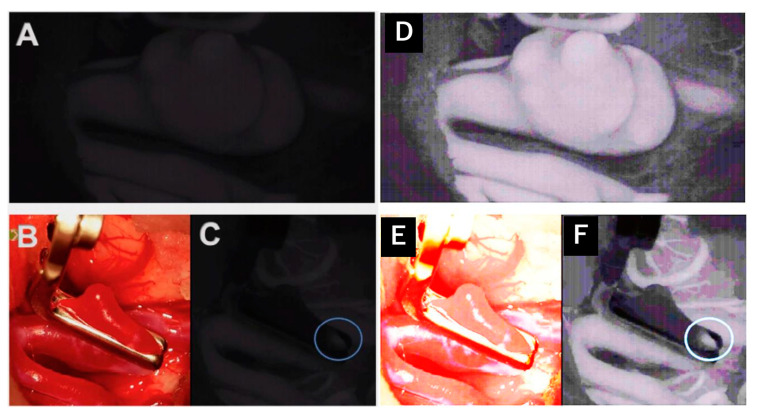
Intraoperative administration of indocyanine green during aneurysm clipping: Perception of indocyanine green during Aeos surgery in A and C. The unclipped aneurysm of the medial cerebral artery is dissected and visualized in (**A**). Visualization of the aneurysm under conventional light during surgery closed with a right-angled Sugita clip in (**B**). In (**C**) the light of the Aeos is on for post clipping control with indocyanine green. Rest perfusion in the clipped sack of the aneurysm at the right angle is highlighted with indocyanine green. Figure (**D**–**F**) show even enhanced contrast by image processing, which however intraoperativly is not necessary due to high definition backlighted LED screen.

**Figure 5 cancers-13-04273-f005:**
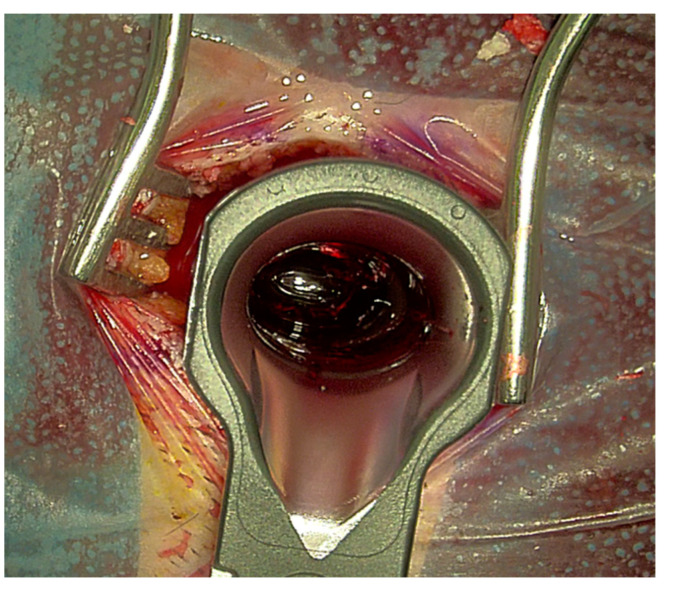
Evacuation of an intracerebral hemorrhage with the assistance of tubes. Intraoperative visualization of the hemorrhage during Aeos surgery.

**Figure 6 cancers-13-04273-f006:**
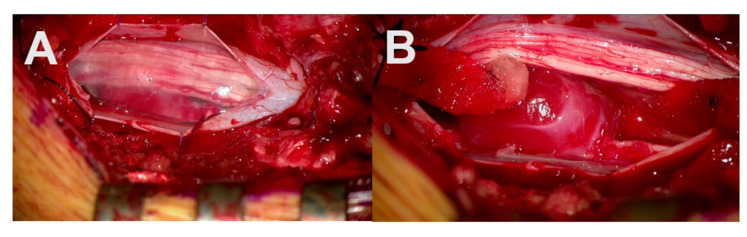
Resection of an intraspinal, lumbar hemangioblastoma with Aeos. Visualization of the arachnoidea and spinal cord after dura opening during surgery in (**A**). Exposure of the underlying intradural tumor in (**B**).

**Figure 7 cancers-13-04273-f007:**
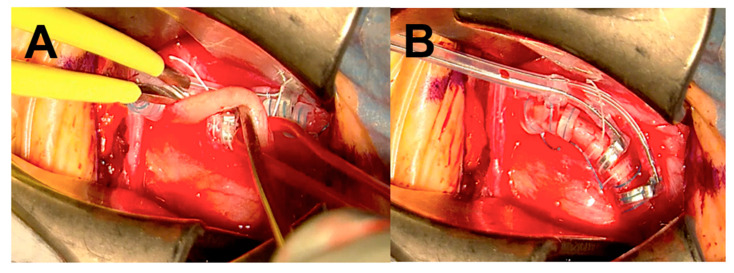
A screenshot of the intraoperative recordings of vagus nerve stimulation in a patient suffering from therapy-resistant depression. The vagus nerve is illustrated in (**A**) during electrode placement. (**B**) represents the final view after electrode placement.

**Figure 8 cancers-13-04273-f008:**
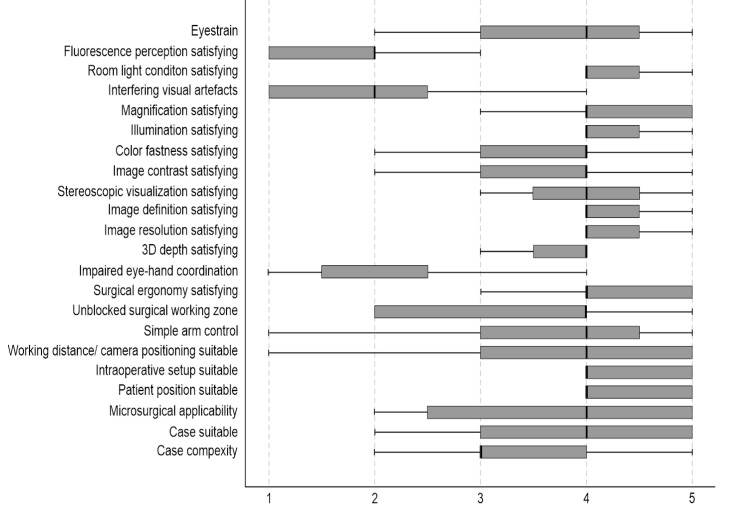
The first questionnaire evaluated the surgeon’s satisfaction regarding the applicability of Aeos. Factors influencing the applicability are listed on the *y*-axis. *X*-axis shows the level of agreement, where 0 is full disagreement, and 5 is full agreement. Boxplots illustrate the median agreement.

**Figure 9 cancers-13-04273-f009:**
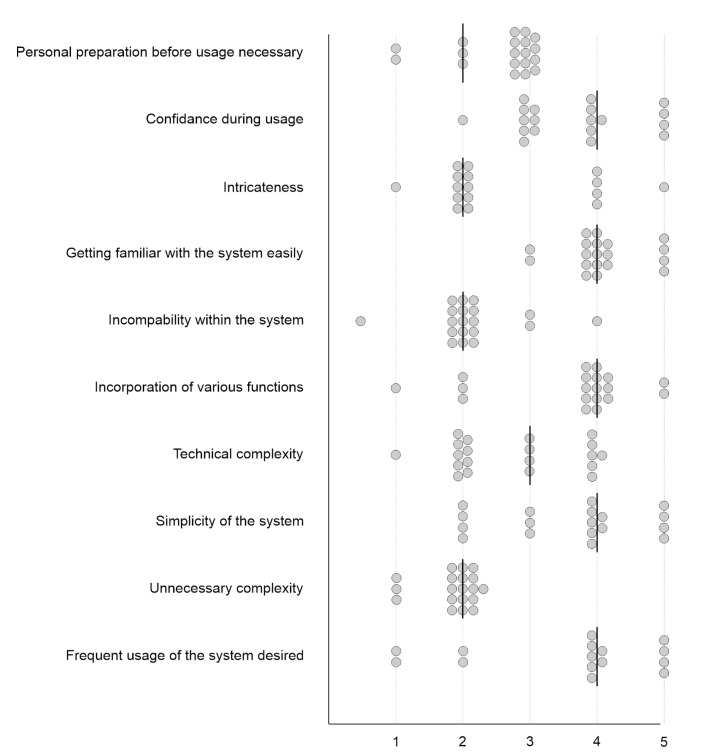
The second questionnaire evaluated the usability of Aeos. The attending surgeon rated the factors listed on the *y*-axis on a scale from 0, and 5 shown on the *x*-axis, where 0 is full disagreement, and 5 is full agreement. Dots illustrate ratings and lines illustrate medians.

**Figure 10 cancers-13-04273-f010:**
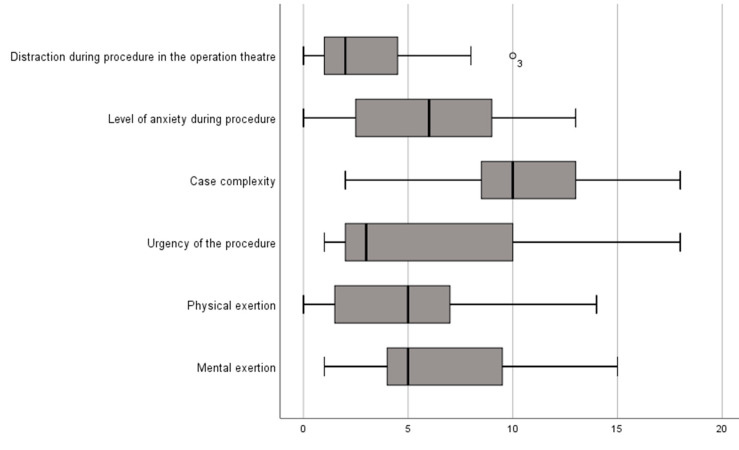
The third questionnaire evaluated the strain of the performing surgeon. The surgeon rated the factors listed on the *y*-axis on a level of agreement between 0 and 20, illustrated on the *x*-axis, where 0 is full disagreement and 20 is full agreement. Boxplots illustrate median agreement.

**Table 1 cancers-13-04273-t001:** The five different types of modern Exoscopes, including technical data.

Platform	ORBEYE	VITOM	MODUS V	KINEVO	AESCULAP Aeos
Company	Olympus, Tokyo, Japan	Karl Storz SE and Co KG, Tuttlingen, Germany	Synaptive Medical, Toronto, Canada	Carl Zeiss AG, Oberkochen, Germany	Braun, Melsungen, Germany
Structure	Exoscope	Exoscope	Exoscope	Exoscope + microscope + endoscope	Exoscope
Additions	Navigation, controller, foot switch	Navigation, hand grips	Voice-activated control	Navigation, hand grips, QEVO	Hand grips, footswitch
Robotic arm	Hands-free, *5*-axis	Pneumatic arm *6*-axis	Hands-free, position memory, *6*-axis	Point-lock, position memory	Log-on-target, waypoints, position memory, *6*-axis
Optics	3D, 4D, 4K	3D, 4D, 4K	2D, HD	3D, 4D, 4K	2D, 3D, HD, 4K
Zoom	26×	8–30×	12.5×	10×	10×
Working distance	220–550 mm	20–50 mm	650 mm	200–625 mm	200–450 mm
Features	ICG, fluorescence, narrow band imaging	ICG	Tractography	ICG, flow assessments, fluorescence	ICG, fluoescence
Light source	LED light, blue light adjustable	LED light, fiberglas conduction,	N/A	N/A	LED light, white/blue light adjustable and simultaneous use

**Table 2 cancers-13-04273-t002:** Procedures listed by localization, procedure type and diagnosis.

Localization	Procedure Type/Diagnosis	Number
Brain		12
	Glioblastoma	3
	Metastasis	3
	Meningiomas	2
	Epilepsy surgery	1
	Aneurysm Clipping	2
	Intracerebral hemorrhage	1
Spinal surgery		6
	Meningioma Th 4/5	1
	Hemangioblastoma L 3/4	1
	Spinal canal stenosis (LW 2/3, C 5-7)	2
	herniated discs L5/S1	2
Peripheral nerve		1
	Vagus nerve stimulation	1

## Data Availability

The data presented in this study are available on request from the corresponding author.

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
