# Peer review of "Evaluation of a Novel Three-Dimensional Robotic Digital Microscope (Aeos) in Neurosurgery"

_cancers, 2021, doi:10.3390/cancers13174273_

Round 1

Reviewer 1 Report

The authors present user-experience for a 3D Robotic Digital Microscope (AEOS) in neurosurgery. 19 surgeons participate to the study and perform different operations at different localizations. The study was conducted by preparing a questionnaire and descriptive statistics were presented for the answers from the users. The study has limitations such as including low number of participants and case studies. On the other hand, the manuscript is written fairly well and provides insights about the new 3D Robotic Digital Microscope in neurosurgery. I have few comments to improve the submitted study.

A picture for the novel 3D Robotic Digital Microscope can be given. Also, a schematic of the system can be given to show how the system is used during a surgery.

The authors can include differences in the answers for different procedures and localizations in a figure.

Were there differences in the answers for different level of education, experience and training of the clinicians? Again the results can be presented in a figure.

Author Response

Response to the Reviewers.

Reviewer 1:

Comments and Suggestions for Authors

The authors present user-experience for a 3D Robotic Digital Microscope (AEOS) in neurosurgery. 19 surgeons participate to the study and perform different operations at different localizations. The study was conducted by preparing a questionnaire and descriptive statistics were presented for the answers from the users. The study has limitations such as including low number of participants and case studies. On the other hand, the manuscript is written fairly well and provides insights about the new 3D Robotic Digital Microscope in neurosurgery. I have few comments to improve the submitted study.

A picture for the novel 3D Robotic Digital Microscope can be given. Also, a schematic of the system can be given to show how the system is used during a surgery.

We thank the reviewer for this suggestion. We added a representative figure to the manuscript including a representative picture of the 3D Robotic Digital Microscope, as well as a picture during the intraoperative use of the system (Figure1 and Figure 2) .

The authors can include differences in the answers for different procedures and localizations in a figure.

In terms of different procedures (e.g. cranial procedures or spinal procedures) the answers from the performing surgeons did not statistically differ.

Were there differences in the answers for different level of education, experience and training of the clinicians? Again the results can be presented in a figure.

Thank you for this valuable remark. Following up to this, we added a whole sequence: 3.3 Younger vs. experienced surgeons in the manuscript.

Reviewer 2:

Comments and Suggestions for Authors

The paper is a basic experiential study of a new exoscope. It has few quantitative measurements except for the survey results. There is opportunity for at least some quantitative measures such as setup time, difference in cost per procedure (i.e. does the system require any expensive consumables etc.), overall cost differential etc. The paper could benefit greatly from specific comments from the Physicians involved especially where technology was not appropriate or somehow deficient or required them to switch to optical microscopes. There is a lack of detail explaining the system itself and how it differs from other similar systems.

Field not filed

Thank you for alerting us to this mistake. „filed“ was changed to „field“

 Does the Aesculap system differ materially from Stortz, Zeiss, Olympus, and Synaptive? A few words on any principal differences would be helpful for the reader.

Thank you for this advice, we added the Table 1 including important technical data comparing the Olympus ORBEYE, the Karl Storz VITOM, the Synaptive Medical MODUS V, the Carl Zeiss KINEVO and the Aesculap Aeos Exoscopes to the manuscript.

Did technicians from the company assist during cases or did physicians use the systems by themselves?

In the beginning a technician did assist during surgery. However, as pointed out in our questionnaires, surgeons got used to the system very fast.

Was setup time different?

Setup time did not differ, compared to the OPMI.

What were these procedures? Please list or explain what were “appropriate” i.e. specific indication or exclusions? Later you indicate that vascular procedures were not suitable but what were the indications for exclusion at the start of the study?

Due to the unknown handling, we started to evaluate the exoscope in standard procedures and therefore excluded very complex cases (ruptured aneurysms, AVM SM °IV or V, forth ventricle ependymomas) and procedures performed in the brainstem (cavernomas).

A photo of the robot showing some of the features would be very useful. How does the ICG fluorescent “backlighting” work?

We thank the reviewer for this suggestion. We added a representative figure to the manuscript including a representative picture of the 3D Robotic Digital Microscope and a picture of a senior physician using the device from the front and from the back (please see above as well).

Is the DICOM for output only or does the system allow for import of CT or MR images and integration into the display?

Yes, it is possible to import data from an MRI or a CT scan into the is possible to Aeos system. We added this feature to the manuscript (2.2 technical specifications).

Table 1: replace “lumbar slipped discs” with “herniated disc”

Thank you for this suggestion. It was changed accordingly.

The Captions A/B/C should be on the respective frames not beside them.

The captions were changed accordingly.

Replace “aneurysma” with “aneurysm”.

Done.

Why were these not considered suitable? This implies that the clipping in Fig 2 was not suitable. Please explain in the caption what the surgeon found unsuitable.

Whereas we felt safe to clip the aneurysm, our different positioning was uncomfortable, in the sense of ‘unnaturally’. Assuming, that a pterional MCA and ICA-aneurysm clipping is one of the most standardized procedure in neurosurgery (instead of tumors, which were located differently), we have to focus to prepare the Sylvian fissure, which was uncomfortable due to the different positioning. Therefore, we would recommend to have a conventional microsocope next to exoscope and to train to use the exoscope. We used it in our fifth case, and handled it well, but not as fast as with a conventional microscope, which differs to the experience in other procedures.

198: The presented image is not from a conventional microscope. This needs to be reworded to say that although this image is from the Aesculap system, it is almost identical to what the physician sees using conventional microscopy.

In this matter, we used a picture in 2D and not in 3D to avoid the blurriness of the picture. Intraoperatively the surgeons used 3D glasses.

215: “In general, neurosurgeons became accustomed to improvable working conditions related to bad ergonomics while performing an operative procedure.”  I’m not sure what this means. Can you please rephrase?

Thank you very much for pointing this out. Indeed, this sentence is hard to understand. We have rephrased the sentence as follows:

“In general, neurosurgeons have become accustomed to restricted intraoperative working conditions causing impaired ergonomics during surgery.”  

218: “ocular dependence” not sure what you mean here.

This refers to being dependent on the OPMI.

220: change “better ergonomical posturing” with “superior ergonomics”

Thank you for pointing this out. We changed “better ergonomical posturing” to “superior ergonomics”

231: Why were the clipping cases not considered suitable? The surgeons need to explain what happened here.

Whereas we felt save to clip the aneurysm, our different positioning was uncomfortable, in the sense of ‘unnaturally’. Assuming, that a pterional MCA and ICA-aneurysm clipping is one of the most standardized procedure in neurosurgery (instead of tumors, which were located differently), we have to focus to prepare the Sylvian fissure, which was uncomfortable due to the different positioning. Therefore, we would recommend to have a conventional microsocope next to exoscope and to train to use the exoscope. We used it in our fifth case, and handled it well, but not as fast as with an conventional microscopes, which differs to the experience in other procedures.

242: “different viewing” is not really a word. What exactly do you mean?

Thank you for alerting us to this. We are referring to focusing on the screen instead of looking through the conventional microscope. To clarify this the have changed the wording to „focussing on the screen“.

256: “the independent working zoom function”. What is this?

When the surgeon moves the camera and stays in place, the independent working zoom function focusses on itself.

257: “intermittently blocked view”. What caused this?

As you can see in the recently inserted Figure 1, the surgeon adjusts the handles to ensure an unblocked view of the screen. This takes some practice and can result in an intermitted blocked view in the beginning.

Did surgeons answer differently when they started and after they were experienced with it?

The surgeons did not answer differently after having more experience with the system.

Reviewer 2 Report

The paper is a basic experiential study of a new exoscope. It has few quantitative measurements except for the survey results. There is opportunity for at least some quantitative measures such as setup time, difference in cost per procedure (i.e. does the system require any expensive consumables etc.), overall cost differential etc. The paper could benefit greatly from specific comments from the Physicians involved especially where technology was not appropriate or somehow deficient or required them to switch to optical microscopes. There is a lack of detail explaining the system itself and how it differs from other similar systems.

34: Field not filed

48: Does the Aesculap system differ materially from Stortz, Zeiss, Olympus, and Synaptive? A few words on any principal differences would be helpful for the reader.

The types of cases vary widely so no consensus can be determined whether the system was comparable to optical microscopes.

Did technicians from the company assist during cases or did physicians use the systems by themselves?

Was setup time different?

63: what were these procedures? Please list or explain what were “appropriate” i.e. specific indication or exclusions? Later you indicate that vascular procedures were not suitable but what were the indications for exclusion at the start of the study?

71: A photo of the robot showing some of the features would be very useful. How does the ICG fluorescent “backlighting” work?

76: Is the DICOM for output only or does the system allow for import of CT or MR images and integration into the display?

117: Table 1: replace “lumbar slipped discs” with “herniated disc”

123: The Captions A/B/C should be on the respective frames not beside them

146: replace “aneurysma” with “aneurysm”

147: Why were these not considered suitable? This implies that the clipping in Fig 2 was not suitable. Please explain in the caption what the surgeon found unsuitable.

198: The presented image is not from a conventional microscope. This needs to be reworded to say that although this image is from the Aesculap system, it is almost identical to what the physician sees using conventional microscopy.

215: “In general, neurosurgeons became accustomed to improvable working conditions related to bad ergonomics while performing an operative procedure.”  I’m not sure what this means. Can you please rephrase?

218: “ocular dependence” not sure what you mean here.

220: change “better ergonomical posturing” with “superior ergonomics”

231: Why were the clipping cases not considered suitable? The surgeons need to explain what happened here.

242: “different viewing” is not really a word. What exactly do you mean?

256: “the independent working zoom function”. What is this?

257: “intermittently blocked view”. What caused this?

Did surgeons answer differently when they started and after they were experienced with it?

Author Response

(The authors gave the same response as above.)

Round 2

Reviewer 1 Report

The authors respond to the reviewer's comments sufficiently. The submitted manuscript can be accepted as a publication.